# Comparative Analysis of Knowledge Control and Evaluation Methods in Higher Education

**Nitza Davidovitch** [1] **, Aleksandra Gerkerova** [1,*] **and Olga Kyselyova** [2]

1    Education Department, Ariel University, Ariel 40700, Israel; d.nitza@ariel.ac.il
2    Department of Quality, Standardization and Metro-logy Odessa, State University of Intelligent Technologies and Communications, 65023 Odesa, Ukraine; kiselovao@ukr.net
\*    Correspondence: gerkerova@ukr.net

**Abstract:** The article analyses knowledge control and evaluation methods in higher education, focusing on both standardized and non-standardized testing. It explores fundamental quality criteria and the perspectives of lecturers and students on the roles of these methods in assessing knowledge. The study evaluates attitudes towards both testing types, considering factors such as efficiency, usability, reliability, objectivity, accuracy, and content relevance. The findings suggest that combining standardized and non-standardized tests with problem-solving tasks significantly enhances knowledge assessment in technical disciplines, demonstrating the methods' interdependence and complementarity. Data shows that both testing types achieve high scores, with standardized tests receiving an integral index of 0.72 and non-standardized tests scoring 0.69. Respondents positively evaluate the effectiveness and convenience of standardized tests, attributing their reliability, objectivity, accuracy, and practical orientation. Although there is a clear preference for standardized tests among participants, the benefits of non-standardized tests are also acknowledged. The study thus recommends a balanced approach, incorporating both methods to ensure an effective and high-quality assessment and knowledge control strategy in higher education.

**Keywords:** standardized test; non-standardized test; knowledge assessment; knowledge evaluation; test quality criteria; content relevance; effectiveness; usability

## 1. Introduction

The modern higher education system adopts a competency-based approach, emphasizing educational mobility and necessitating a comprehensive assessment of knowledge. The prevalent method for monitoring and assessing knowledge in higher education is testing, recognized for its quality, objectivity, speed, and efficacy [1]. Contemporary higher education institutions prioritize treating students as active participants in their learning and development, valuing student feedback on teaching methodologies [2]. Students are expected to independently acquire knowledge, formulate opinions, apply concepts in various contexts, and exhibit self-development, learning, and realization. These evolving requirements prompt the exploration of new assessment approaches, which must encompass all presented educational knowledge, be adaptable to various training forms, and demonstrate evidence-based effectiveness.

The use of high-quality methods to assess students' knowledge and skills is crucial. Test-based assessment has emerged as the dominant pedagogical measurement approach. Research indicates that combining standardized test materials with traditional teaching and assessment methods is crucial to enhancing educational quality. A standardized test, as defined by Nathan R. Kuncel and Sarah A. Hezlett [3], is one that has undergone pilot testing, exhibits stable and acceptable quality metrics, and has approved instructions for repeated use. Examples include internationally recognized English language proficiency tests (e.g., TOEFL, BEC) and multidisciplinary university admission tests (e.g., SSAT, SAT,

ACT, GRE, GMAT) in the USA and Europe, alongside the Ukrainian National Multi-Subject Test—External Independent Testing and Matriculation exams (Bagrut) in Israel. Standardized tests, such as those for language proficiency and university admissions, are prevalent across many European countries. Having undergone years of validation, these tests have proven effective [4]. However, it is important to recognize that due to the variability and specialized nature of university disciplines, standardized tests for every subject do not exist. In these instances, educators resort to creating non-standardized tests. These tests undergo essential methodological procedures to ensure their reliability, validity, and objectivity, albeit they might not meet the rigorous standards of standardization. Despite the prominence of test-based assessment, especially in science and engineering, its capability to fully evaluate student competencies and the feasibility of developing quality tests at universities warrant scrutiny [5]. The educational community's attitudes toward the widespread adoption of test-based control remain a critical area of interest.

## 2. Research Questions

1. What are the essential quality criteria for both standardized and non-standardized tests?
2. According to teachers and students, which tests are perceived to be of the highest quality?
3. Can non-standardized tests be effectively utilized and potentially replace standardized tests? If so, what implications would this have on the quality of knowledge assessment?

## 3. Literature Review

The modern testing system, particularly for multidisciplinary tests, is grounded in the principle of assimilation elements—Bloom's taxonomy [6], which suggests that tests should cover knowledge, comprehension, application, analysis, synthesis, and evaluation. However, Tonya R. Moon et al. have identified a critical issue: in aiming for the highest scores, students often focus solely on test preparation, which can lead to a disconnect from the learning process. This narrow focus may undermine the understanding of cause-and-effect relationships, logical reasoning, and the synthesis of new knowledge, thereby diminishing the quality of knowledge and its practical application. Their analysis reveals that while knowledge and comprehension are effectively assessed, the application, analysis, synthesis, and evaluation components often remain untested [7].

Research by Kuncel and Hezlett supports the notion that standardized tests can provide valuable insights into the quality of acquired knowledge and predict future learning outcomes. However, they argue for the integration of various non-standard assessment methods, such as problem-solving, creative, and professionally oriented tasks, to boost student motivation and prompt further research in this field [3].

Professor Rawlusyk advocates for a diverse assessment approach to ensure quality knowledge assessment, incorporating methods like tests, authentic learning tasks, student self-assessment, collective assessment, and feedback to gauge learning potential. Despite the challenges associated with authentic methods and testing, Rawlusyk emphasizes the importance of balancing these approaches to maintain learning quality. His findings highlight the limited student involvement in selecting control strategies and assessment methods, underscoring the need for further research to explore this issue [8].

The modern higher education system, grounded in a competency-based approach, necessitates a similarly foundational approach in the development of assessment tests. Pavlina Krešimir et al. posit that existing tests typically measure only a singular type of competence, underscoring the need for tests that evaluate interdisciplinary and broader competencies, including personal, social, professional, and communication skills. This approach necessitates the incorporation of practical, logical, and analytical problem-solving into testing [9]. In this context, Peg Tyre's research is particularly pertinent. Focusing on the comprehensive implementation of standardized tests in U.S. colleges, Tyre observes that while many institutions have embraced standardized testing as effective, there remains a contingent, notably within private universities, advocating for the creation of bespoke tests. These tests would better reflect the unique characteristics of the institution and the

students' knowledge levels, arguing that standardized tests may not fully capture the variety, breadth, and depth of discipline-specific knowledge and learning present in today's colleges and universities [10].

A review of the literature reveals a lack of consensus within the academic community regarding the efficacy of standardized tests, leaving open the question of the necessity for integrating non-standardized tests into the educational process—a question that has become central to our study.

Further analysis by scholars [11–14] provides a foundation for asserting that the quality of testing is contingent upon criteria that may fluctuate based on the specific objectives and conditions of the test administration. Identified widely accepted criteria include the alignment of test content with testing objectives, reliability, objectivity, scientific rigor, validity, differentiation, standardization, variability, and consistency. These characteristics have informed the definition of quality criteria for standardized and non-standardized tests in our study: "efficacy", "usability", "reliability", "objectivity", "accuracy", and "content relevance".

During the study, respondents were not asked to assess criteria such as test validity, scientific soundness, and standardization due to the conceptual challenges and interpretative ambiguities encountered by students and teachers alike. To elucidate, validity concerns the degree to which a test accurately measures the intended quality and how well the test results reflect objective reality [13]. Scientificality entails the inclusion of scientific concepts within the test [14]. Standardization denotes the uniformity of test administration and assessment procedures [15].

To facilitate a comprehensive understanding among respondents, we provided concise definitions of the key concepts employed in our subsequent survey: efficacy, usability, reliability, objectivity, accuracy, and content relevance.

Efficacy of a standardized test is measured by its ability to yield high-quality results with the fewest possible tasks, offering individualized assessment for each test-taker by optimizing complexity and minimizing task quantity [12].

Usability of a standardized test encompasses a standardized and regulated procedure for administering, processing, and analyzing test results. Key aspects of usability include unambiguous, clear instructions, accessibility, a convenient format, a reasonable time for completion, a difficulty level matching student preparedness, clear structure, the possibility of revisiting questions, and provision of feedback [14].

Reliability refers to the consistency and precision of a test in evaluating the specific characteristics or skills. It indicates the stability of test results when administered repeatedly under identical conditions (test–retest reliability). Internal consistency concerns the uniformity of test items concerning the trait measured, suggesting each item contributes towards assessing the intended quality [14].

Objectivity in testing is the extent to which test results reliably and accurately represent students' competencies, knowledge, abilities, and skills, unaffected by subjective biases, achieved through standardized procedures [1].

Accuracy of a standardized test is a composite measure determined by several factors: reliability, validity, standardization, monitoring, auditing, task variety, and statistical indicators. These elements collectively assess the precision of a standardized test, enhancing its effectiveness in evaluating student knowledge and skills [16].

Content Relevance assesses the extent to which test tasks accurately mirror the educational program's material. This criterion can be quantified using the following formula:

$$S = \frac{N_p}{N_t}$$

Here, $N_p$ represents the number of descriptors (basic concepts, terms, formulas, etc.) found within the educational program's thesaurus, and $N_t$ signifies the number of descriptors present in the test thesaurus. An educational thesaurus comprises a set of descriptors and the relationships between them, serving as a structured representation of the pro-

gram's knowledge domain [14]. The ratio *S* thus provides a direct measure of the degree to which the test content aligns with the educational objectives and curriculum, ensuring the assessment's relevance to the taught material.

Standardized tests are characterized by well-defined criteria that outline the key features of test control, facilitating quality analysis based on these criteria. In contrast, non-standardized tests lack precise characteristics and scoring criteria, complicating the assessment of their quality. Non-standardized tests, often traditional in approach, prioritize practical application, assessing not only knowledge but also the ability to apply it in practical contexts, such as demonstrating acquired skills. In technical disciplines, a well-constructed non-standardized test includes tasks of varying complexity, solution methods, and formats. Typically, these tests present questions and tasks in a progressive sequence, starting with simpler tasks and advancing to more challenging ones. At the test's conclusion, students may encounter additional problems or assignments of higher difficulty, offering the chance to gain extra points for correct completion.

The analysis of scientific literature has allowed us to identify the principal criteria that characterize both standardized and non-standardized tests. This understanding of the key distinctions between standardized and non-standardized tests has shaped the direction and scope of our research.

## 4. Research Methods

Our study employed a blend of theoretical and empirical methods to address the research questions. The theoretical framework encompassed a variety of analytical approaches, including analysis, synthesis, comparison, systematization, forecasting, and generalization of information pertinent to the research issue. Additionally, це employed the method of secondary information processing.

On the empirical front, data collection was conducted through surveys and interviews, providing firsthand insights from participants. Statistical methods were instrumental in analyzing the data, employing descriptive statistics for data presentation (including tables, graphical representations, and quantification) and the theory of experimental design to identify and evaluate cause-and-effect relationships among variables.

To analyze it, we focused on the following standardized tests: the National Ukrainian Multi-Subject Test (EIT), tests in higher mathematics, and tests in electrical engineering and electronics, which are administered across Ukrainian universities that participated in our survey [17,18]. Additionally, we examined non-standardized tests in various disciplines, specifically developed and utilized by lecturers at a particular university, including higher mathematics, information and measurement systems, metrology, electrical engineering, and the strength of materials.

## 5. Research Results

The study aimed to explore the relationship between the quality of knowledge assessment methods and their perception by lecturers and students in technical courses, focusing on their attitudes towards standardized and non-standardized tests. This was to evaluate the tests' quality and effectiveness within the university's educational process. A survey was conducted among the academic populations of four universities located in Odessa, Ukraine, and Ariel, Israel, at the State University of Intelligent Technologies and Telecommunications (42 students and 18 lecturers), Odessa State Academy of Construction and Architecture (46 students and 24 lecturers), National University "Odessa Polytechnic" (38 students and 24 lecturers), and Ariel University (112 students and 32 lecturers).

The survey encompassed 238 students and 98 lecturers, evaluating the quality of both standardized tests and non-standardized tests designed by lecturers for the disciplines studied. The tests analyzed included multiple-choice questions with three to five options, along with items requiring detailed written responses and problem-solving.

The student respondents were analyzed based on several criteria to capture a comprehensive demographic and experiential profile: country (to identify intercultural differences),

year of study (to assess the impact of educational experience on survey responses), gender (to explore potential differences in responses between genders), and age (to examine the influence of age, and by extension, life experience, on their responses). The findings are detailed in Table 1.

**Table 1.** Demographic and Educational Profile of Student Respondents.

| Country | Year of Study | Gender | Age |
|---|---|---|---|
| Ukraine—126 students | 1–2—44 students<br>3–4—47 students<br>Magistracy—35 students | m. 72<br>f. 54 | 19–21 years old—59<br>22–25 years old—47<br>26+ years old—20 |
| Israel—112 students | 1–2—46 students<br>3–4—38 students<br>Magistracy—28 | m. 60<br>f. 52 | 19–21 years old—32<br>22–25 years old—49<br>26+ years old—31 |

Our analysis revealed that the variance in responses between participants from Ukraine and Israel was negligible, with a deviation of only 0.1%. Furthermore, the investigation did not identify any significant correlations between the obtained data and the respondents' age, gender, or course of study. This suggests that perceptions of test quality and efficacy in the educational process are consistent across these demographic and academic variables.

In addition to the student cohort, we conducted a similar analysis on the group of lecturers involved in the survey. The lecturers were evaluated based on criteria such as country of employment, years of teaching experience, and age. The findings from this segment of our study are detailed in Table 2.

**Table 2.** Overview of Lecturer Demographics and Experience.

| Country | Teaching Experience | Gender | Age |
|---|---|---|---|
| Ukraine—98 | 1–5 years—11<br>6–12 years—28<br>13–20 years—32<br>more than 20 years—27 | m. 51<br>f. 47 | 30–38 years old—18<br>39–50 years old—34<br>50+ years old—14 |
| Israel—32 | 1–5 years—6 6–12 years—11<br>13–20 years—11<br>more than 20 years—4 | m. 14<br>f. 18 | 30–38 years old—4<br>39–50 years old—21<br>50+ years old—7 |

Similar to the student cohort, our analysis of the lecturers' responses revealed marginal differences (0.2%) between participants from Ukraine and Israel. Additionally, no significant trends were observed concerning the responses' dependency on the lecturers' work experience, gender, or age.

From the collective analysis of both student and lecturer samples, we deduced that the minor disparities noted do not hold significant weight within the context of our study and are unlikely to influence its outcomes.

In alignment with contemporary university practices, the ECTS (European Credit Transfer and Accumulation System) grading scale was employed to evaluate student achievements. To maintain consistency across evaluations, non-standardized tests were graded using the same criteria as their standardized counterparts. Prior to the survey's administration, both lecturers and students were briefed on these criteria to preclude any misunderstandings related to terminology.

Participants in the survey included lecturers specializing in natural and technical disciplines, such as higher mathematics, information and measurement systems, metrology, strength of materials, electrical engineering, and electronics. The analysis encompassed standardized tests in higher mathematics, electrical engineering, and electronics alongside results from the multidisciplinary test (EIT) taken by all university applicants in Ukraine. It also included non-standardized tests in disciplines crafted and administered by lecturers at

specific universities. To gauge the quality of these tests, respondents were asked to use a 5-point scale, the details of which are delineated in Table 3.

**Table 3.** Criteria for Assessing Test Quality.

| Points | Meaning |
|---|---|
| 5 | The test fully meets the researched criterion and test requirements, ensuring high-quality knowledge assessment. |
| 4 | The test mostly meets the researched criterion and test requirements, providing sufficiently high-quality knowledge assessment. |
| 3 | The test meets only some characteristics of the researched criterion and basic test requirements, resulting in insufficient quality of knowledge assessment. |
| 2 | The test meets a few characteristics of the criterion and some test requirements, not ensuring the quality of knowledge assessment. |
| 1 | The test does not meet the researched criterion; it is replaced by a disorganized list of questions, lacks test characteristics and structure, and cannot be used as a knowledge assessment tool. |

The findings from the survey conducted among lecturers at universities in Odessa and Ariel are depicted in Figures 1 and 2.

The outcomes of the survey involving students from Odessa universities and Ariel, which sought to ascertain their views on knowledge assessment and control via standardized and non-standardized tests, along with the criteria for their quality, are depicted in Figures 3 and 4.

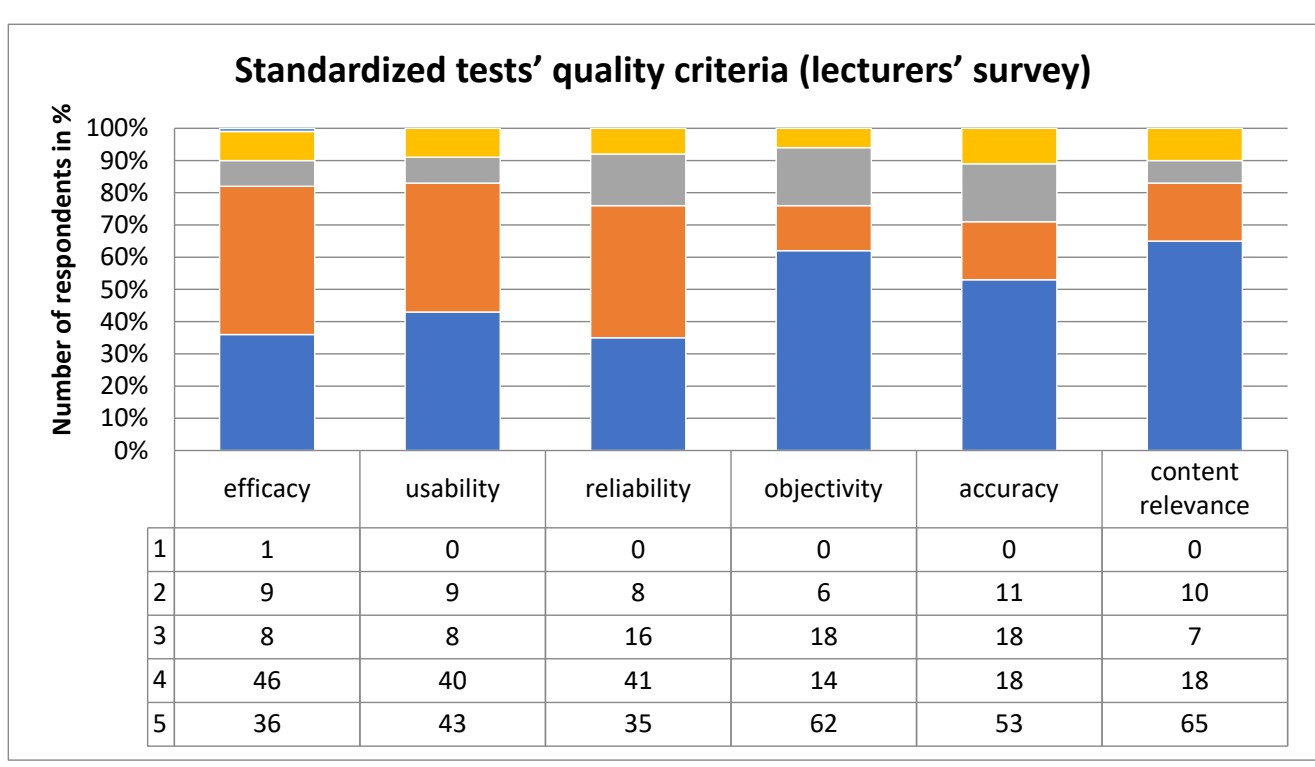

**Figure 1.** Standardized tests' quality criteria (lecturers' survey). 1 point is denoted by ⬜, 2 points by 🟨, 3 points by ⬜, 4 points by 🟧, 5 points by 🟦.

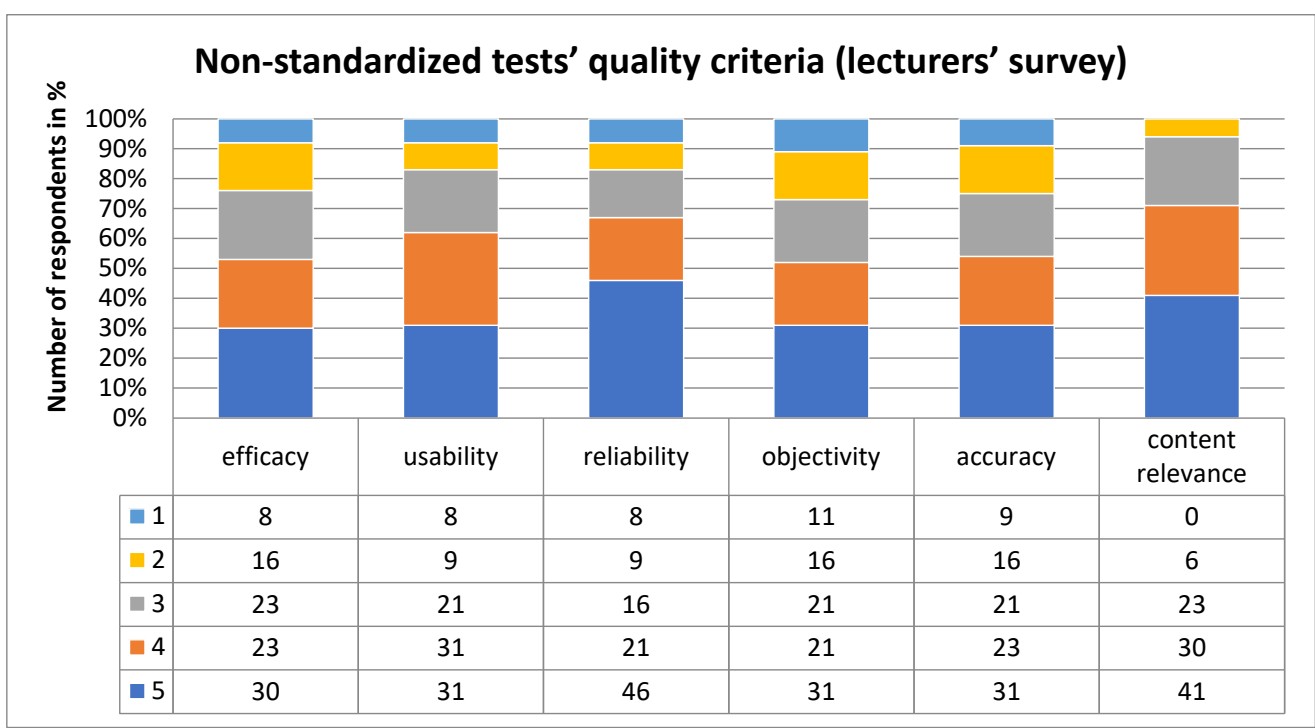

**Figure 2.** Non-standardized tests' quality criteria (lecturers' survey). Note: All survey results are presented as percentages, rounded to whole numbers for clarity. 1 point is denoted by ▢, 2 points by ▢, 3 points by ▢, 4 points by ▢, 5 points by ▢.

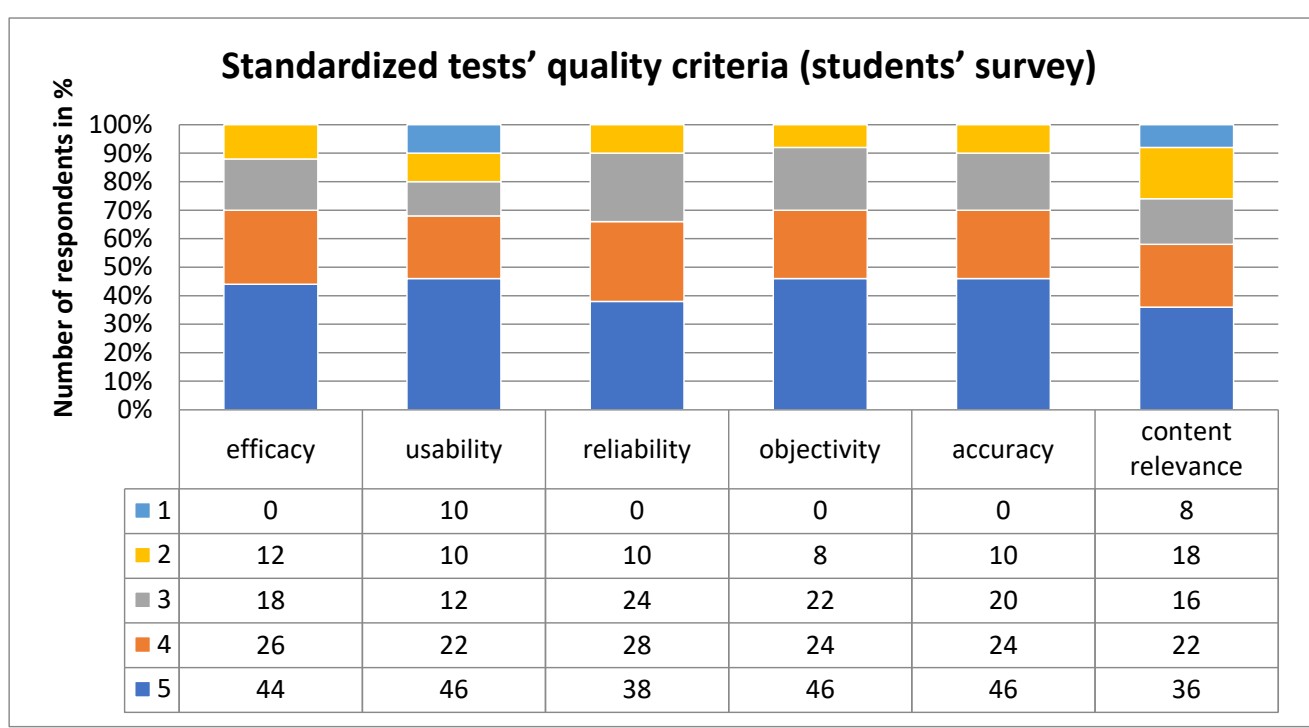

**Figure 3.** Standardized tests' quality criteria (students' survey). 1 point is denoted by ▢, 2 points by ▢, 3 points by ▢, 4 points by ▢, 5 points by ▢.

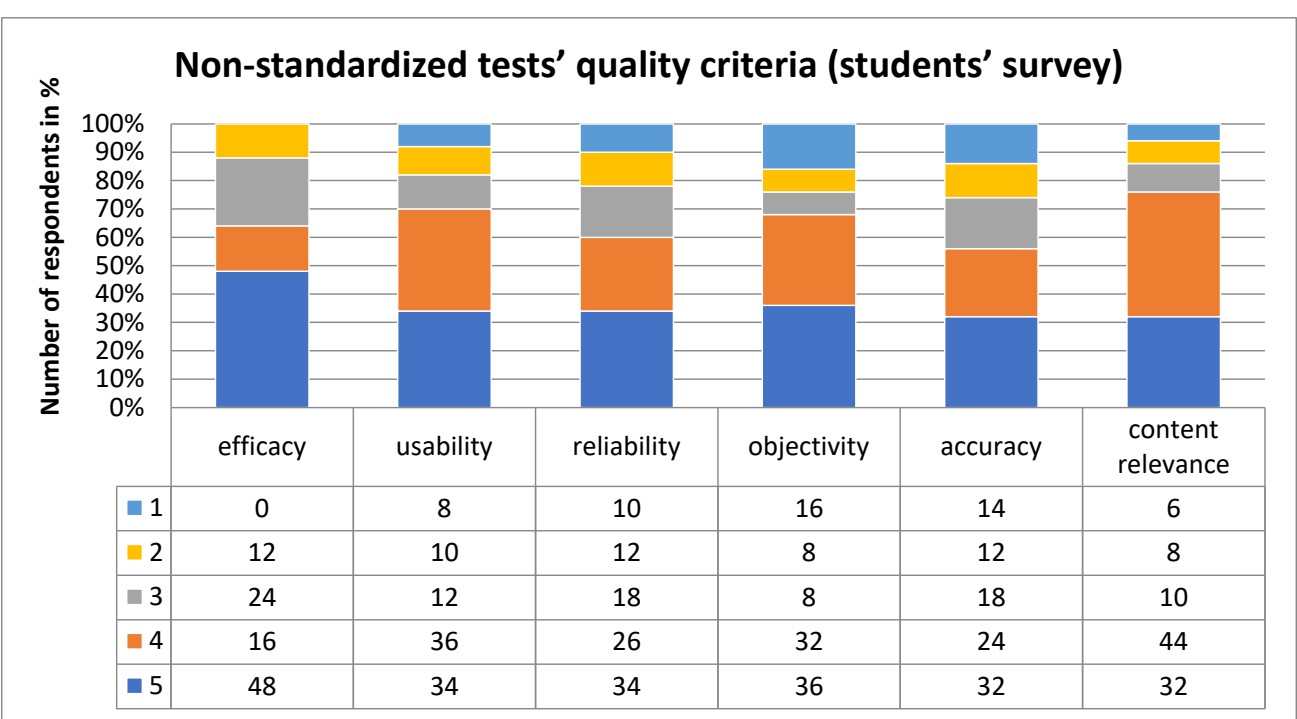

**Figure 4.** Non-standardized tests' quality criteria (students' survey). 1 point is denoted by [   ], 2 points by [   ], 3 points by [   ], 4 points by [   ], 5 points by [   ].

The analysis of the survey results among lecturers and students indicates that both groups have similar evaluations of the quality of knowledge assessment methods across various criteria. Notably, a significant proportion of lecturers (36%) and students (44%) awarded the highest score of 5 points to the efficacy of standardized tests. This criterion was rated 4 points by 46% of lecturers and 26% of students, while 3 points were chosen by 8% of lecturers and 18% of students. A smaller percentage identified the inefficacy of standardized tests, with only 9% of lecturers and 12% of students marking them as 2 points; a minimal 1% of lecturers provided a rating of 1 point. Lecturers highlighted that standardized tests efficiently assess students' knowledge of specific topics. Students favored standardized tests for ensuring fairness, requiring relatively shorter preparation time compared to other assessment forms, and offering transparent criteria for assessment.

Upon evaluating the "usability of a standardized test", the majority (43% of lecturers and 46% of students) assigned it 5 points; 40% of lecturers and 22% of students rated it 4 points. Scores of 3 points were given by 8% of lecturers and 12% of students, 2 points by 9% of lecturers and 10% of students, and 1 point by 10% of students. The respondents appreciated the standardized test's convenience, attributed to its clear instructions and defined assessment criteria.

For the "reliability of a standardized test", 32% of lecturers and 38% of students gave it 5 points. A 4-point rating was given by 41% of lecturers and 28% of students; 16% of lecturers and 24% of students assigned it 3 points. Only 8% of lecturers and 10% of students rated the test's reliability as 2 points. Notably, no participant considered the standardized test completely unreliable, indicating a consensus on their reliability due to standardized procedures and preliminary testing on a large sample.

The "objectivity" criterion further showed widespread approval for standardized tests: 62% of lecturers and 46% of students gave 5 points, and 14% of lecturers and 24% of students gave 4 points. Some respondents expressed doubts about the objectivity of standardized tests, with 6% of lecturers and 8% of students marking 2 points. Yet, no one considered these tests biased, highlighting their objective nature due to broad-based testing and independence from the test compiler's professional skills.

For the "test accuracy" criterion, standardized tests received a rating of 5 points from 53% of lecturers and 46% of students, with 18% of lecturers and 24% of students assigning 4 points, and 18% of lecturers alongside 20% of students giving 3 points. A minority, 11% of lecturers and 10% of students, questioned the accuracy of standardized tests with a 2-point rating, but no respondents deemed them completely inaccurate (1 point). Feedback suggests that standardized tests are viewed as highly accurate and valuable for gauging acquired knowledge, forecasting learning progression, and facilitating feedback and self-assessment among students.

Regarding "relevance", 65% of lecturers and 36% of students found standardized tests fully relevant (5 points), although 18% of lecturers and 22% of students rated it at 4 points, and a smaller portion, 7% of lecturers and 16% of students, considered it moderately relevant (3 points). Some uncertainty about relevance was expressed by 10% of lecturers and 18% of students (2 points), and a minimal 8% of students found these tests to be not relevant at all. While recognizing the relevance of standardized tests, critics argue that they often evaluate superficial knowledge, such as memorization of formulas, without testing problem-solving applications.

Participants acknowledged the high standards of standardized tests across all criteria, yet pointed out their limited scope in technical fields and the challenges they present in assessing practical, professionally oriented skills, and creativity. Emotional challenges associated with test-taking, like anxiety, were also highlighted by students.

Non-standardized tests' efficacy received a 5-point rating from 30% of lecturers and 48% of students, indicating a slightly lesser efficacy compared to standardized tests attributed to difficulties in structuring educational content. About 23% of lecturers and 24% of students view these tests as somewhat effective but lacking in quality assessment depth.

The "usability" of non-standardized tests was highly rated (5 points) by 31% of lecturers and 34% of students, but concerns were raised about their structure and the overload of answer options. The "reliability" and "objectivity" of non-standardized tests were questioned, with respondents noting potential biases due to the subjective nature of test creation and administration.

For the "reliability" criterion, 46% of lecturers and 34% of students awarded the highest score of 5 points, indicating strong confidence in this aspect. Additionally, 21% of lecturers and 26% of students gave it 4 points, suggesting good but not perfect reliability. A moderate level of reliability, represented by 3 points, was observed by 16% of lecturers and 18% of students. Meanwhile, 9% of lecturers and 12% of students perceived only partial reliability, scoring it 2 points. A small proportion, 8% of lecturers and 10% of students, perceived the tests as completely unreliable, giving them 1 point. The feedback highlighted concerns over the non-standardized tests' inconsistency and the accuracy in evaluating specific characteristics or skills, underlining potential issues with their reliability.

Regarding the "objectivity" of non-standardized tests, the survey revealed that 31% of lecturers and 36% of students awarded 5 points; 21% of lecturers and 32% of students gave 4 points; 3 points were assigned by 21% of lecturers and 8% of students; 16% of lecturers and 8% of students selected 2 points; and 11% of lecturers along with 16% of students expressed a lack of trust in the objectivity of non-standardized tests (1 point). When compared to standardized tests, non-standardized tests were perceived to have lower objectivity, a sentiment more pronounced among student respondents. This perception is attributed to the variability in test creation and administration, with lecturers sometimes deviating from standardized methods, potentially leading to tests being unnecessarily complicated or overly simplified.

For the "accuracy" criterion, 31% of lecturers and 32% of students awarded 5 points, while 23% of lecturers and 24% of students gave 4 points. A score of 3 points was chosen by 21% of lecturers and 18% of students, and 2 points by 16% of lecturers and 12% of students, while 9% of lecturers and 14% of students assigned the lowest rating of 1 point. These evaluations suggest that non-standardized tests are perceived to have lower accuracy

compared to standardized tests, attributed to the use of imprecise wording and vague concepts, which may introduce assessment bias.

Regarding the "relevance" of non-standardized tests, 41% of lecturers and 32% of students rated them with the highest score of 5 points. Thirty percent of lecturers and a notable 44% of students felt they deserved 4 points. Scores of 3 points were given by 23% of lecturers and 8% of students, while 2 points were selected by 6% of lecturers and 8% of students. A minimal 6% of students perceived these tests as not relevant, awarding 1 point. This high regard for the relevance of non-standardized tests is explained by their alignment with the curriculum, as the tests are typically developed by the instructors responsible for the course, allowing for a tailored assessment that incorporates the full breadth of material based on its significance and the complexity of the topics covered.

In order to delineate the variables of quality criteria effectively, we established a quality assessment scale. This scale incorporates both complex quality criteria (CQC) and an integral quality criterion (IQC), with the grading methodology based on the interpretation of the ECTS grading scale. For the computation of a complex quality criterion, we employed the following formula, which integrates the significance and score of each quality indicator:

$$K = \sum_{i=1}^{n} W_i \cdot P_i$$

where:

$K$ represents the complex quality criterion,

$n$ is the total number of quality indicators considered,

$W_i$ denotes the weight (importance) assigned to the $i$-th quality indicator,

$P_i$ signifies the score of the $i$-th quality indicator.

This approach allows for a nuanced analysis of test quality by accounting for the varying significance of different quality dimensions.

Note: According to the characteristics of the quality scale we have defined (unsatisfactory, satisfactory, good, excellent) and the scores assigned to them (Table 4), the value of the quality scale and the quality criterion are taken from W. Tsyba [19]

**Table 4.** Significant Characteristics of the Quality Assessment Scale.

| Quality Level | Points | Range | Quality Scale Value | Quality Criterion Value |
|---|---|---|---|---|
| Unsatisfactory | 1, 2 | 0–59 | $0 < q < 0.6$ | 0.3 |
| Satisfactory | 3 | 60–74 | $0.6 \leq q < 0.75$ | 0.67 |
| Good | 4 | 75–89 | $0.75 \leq q \leq 0.89$ | 0.86 |
| Excellent | 5 | 90–100 | $0.90 \leq q \leq 1$ | 0.98 |

Using the formula, comparative histograms depicting the values of quality criteria for both standardized and non-standardized tests were created for a sample of lecturers and students (Figures 5 and 6).

From the analysis of the histogram data, it is evident that both respondent groups—lecturers and students—show a preference for standardized tests. The "efficacy" indicator was rated at 0.72 for standardized tests (ST) and 0.66 for non-standardized tests (NST) by lecturers, while students rated it at 0.67 ST and 0.65 NST. The "usability" was evaluated by lecturers as 0.78 ST and 0.61 NST; the disparity is more pronounced among students, with ratings of 0.76 ST and 0.56 NST. For the "reliability" criterion, lecturers rated it at 0.71 ST and 0.76 NST, and students rated it at 0.68 ST and 0.63 NST. The "objectivity" criterion received a numerical value of 0.77 ST and 0.56 NST from lecturers and 0.68 ST and 0.61 NST from students. According to the "accuracy" criterion, lecturers and students both provided a rating of 0.72 ST and 0.63 NST for teachers and 0.72 ST and 0.61 NST for students. The "relevance" value, as per the lecturer survey, is 0.74 ST and 0.65 NST, and for students, it is 0.69 ST and 0.62 NST.

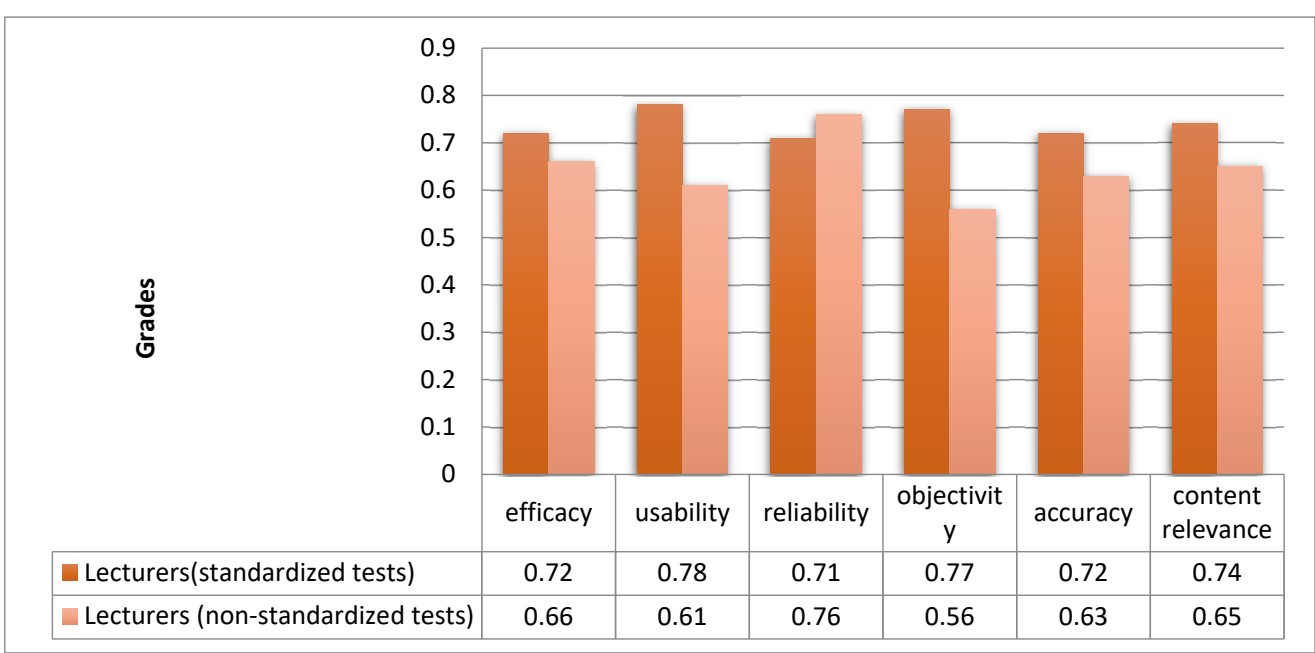

**Figure 5.** Comparative Analysis of Quality Criteria Values (Survey Results from Lecturers).

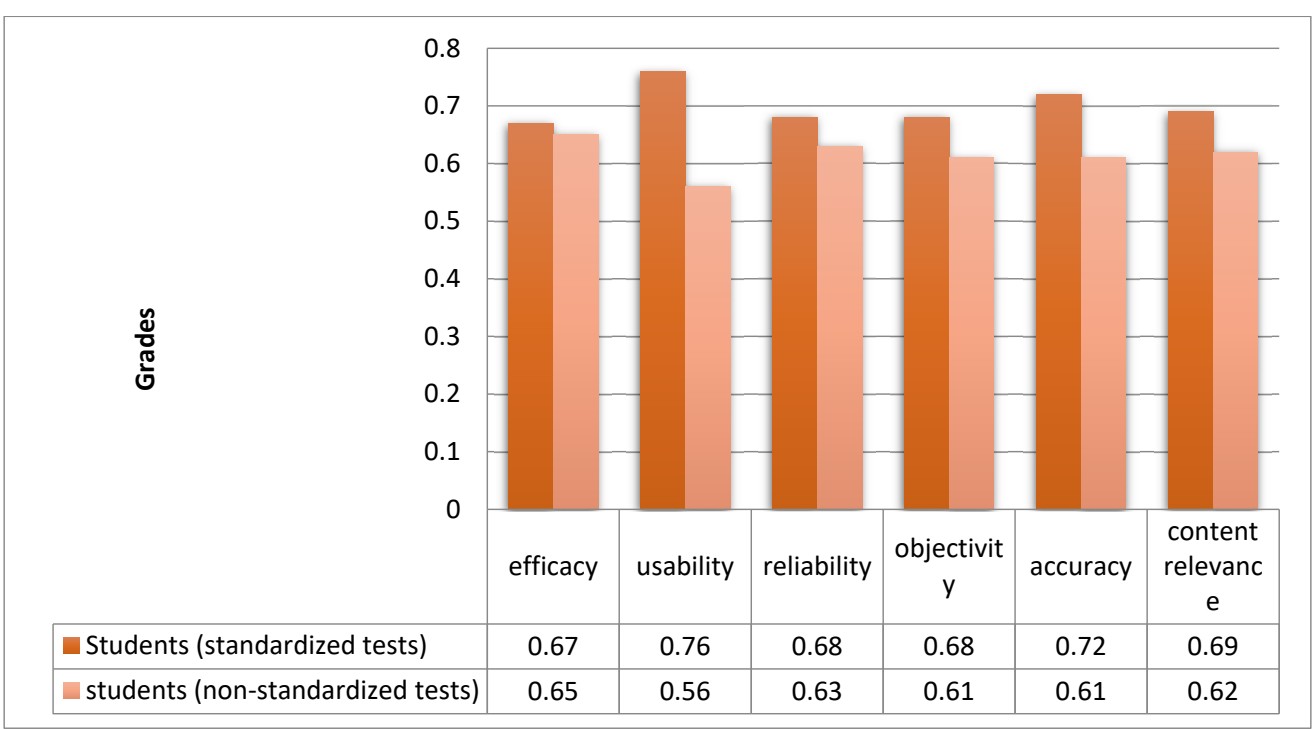

**Figure 6.** Comparative Analysis of Quality Criteria Values (Survey Results from Students).

Using the data provided, a comprehensive quality criterion was calculated (illustrated in Figure 7), indicating a general preference for standardized tests (ST) in knowledge assessment, although the difference is marginal (0.09 for the lecturer sample and 0.08 for students). Lecturers assessed the quality of ST at 0.74 and NST at 0.65, while students provided slightly lower evaluations: 0.7 for ST and 0.62 for NST.

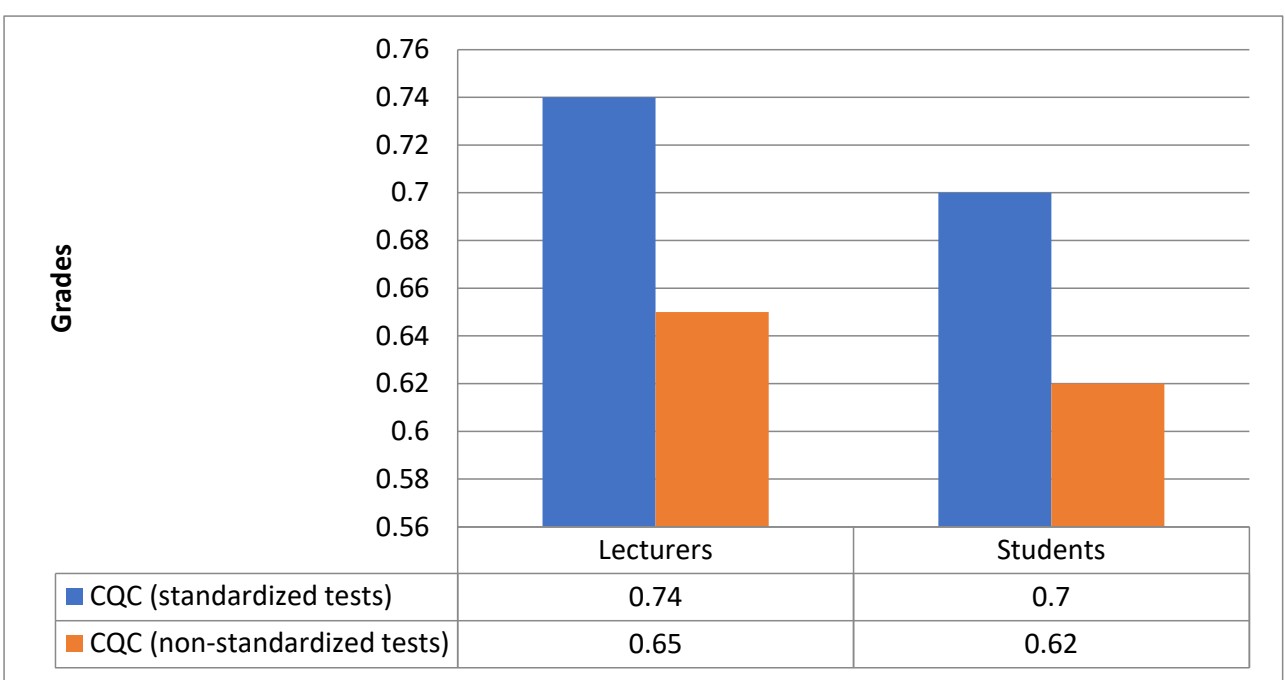

**Figure 7.** Comparative histogram of complex quality criteria.

Based on the results obtained in determining integrated quality criteria (QQC), an integral quality criterion (IQC) was established. The IQC was calculated using the formula:

$$Q_i = \sum_{i=1}^{n} \omega_i \sqrt[n]{\prod_{i=1}^{n} q^{\omega_i}}$$

where $Q_i$ denotes the *i*-th quality criterion comprising the IQC and indicates the weighting coefficient of the *i*-th criterion.

As shown in Figure 8, the data suggest that both teachers and students favor the use of standardized test control as a method for assessing knowledge, despite a minimal difference of 0.08 points in the Integral Quality Criteria (IQC) between standardized and non-standardized tests (ST—0.72 and NST—0.64 points). All participants expressed a positive view regarding the standardization of test evaluation and simultaneously acknowledged the necessity of employing non-standardized tests. These tests, with proper design and a regulated conducting procedure, can ensure quality control and assessment of knowledge for each specific discipline taught at the university. Consequently, it is concluded that a combination of these two methods should be employed for knowledge control and assessment in higher education.

Based on the collected data, it is evident that both lecturers and students show a preference for utilizing standardized tests as a method for knowledge assessment despite a marginal difference of 0.08 points in the integral quality criteria (IQC) between standardized and non-standardized tests (ST—0.72 and NST—0.64 points). Participants uniformly endorsed the standardized approach to test assessment while also acknowledging the importance of incorporating non-standardized tests. They highlighted that, with appropriate preparation and a regulated procedure for administration, non-standardized tests could facilitate high-quality control and assessment of knowledge specific to each discipline taught at the university. Consequently, the findings suggest that an optimal approach to knowledge evaluation in higher education would involve a strategic combination of both standardized and non-standardized testing methods.

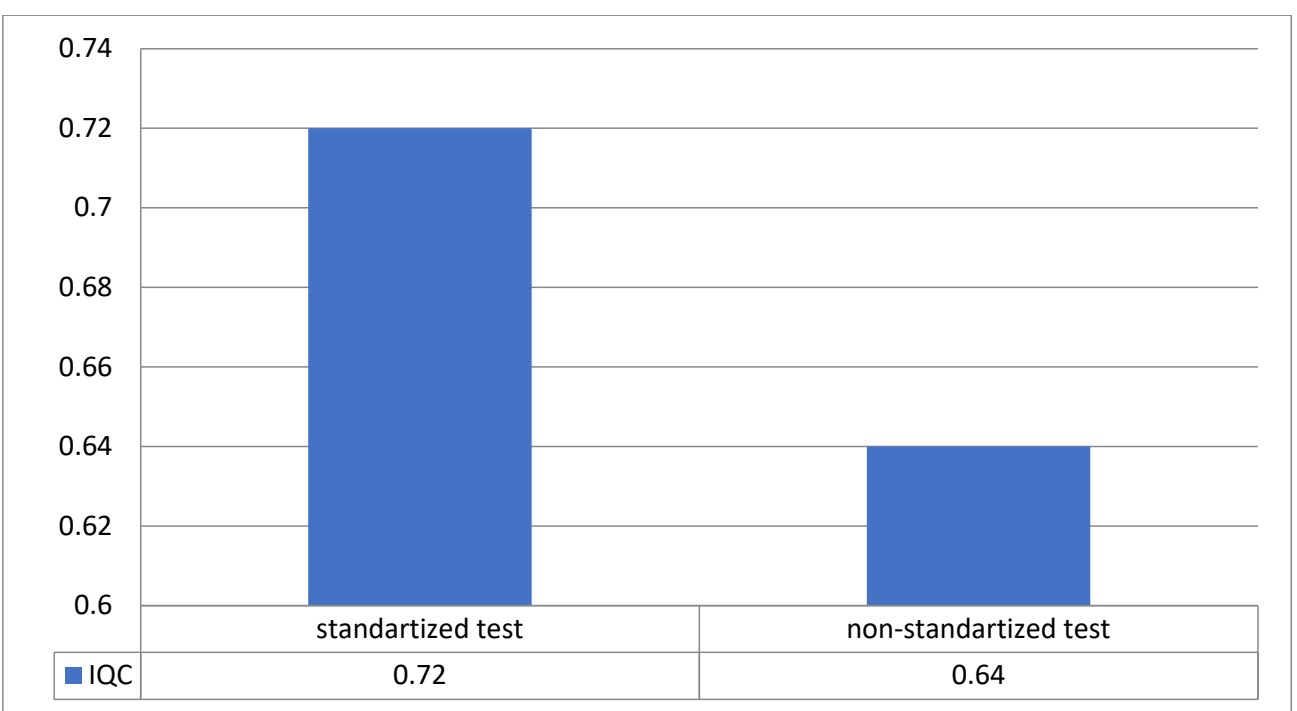

**Figure 8.** Integral quality criterion.

## 6. Discussion

The adoption of standardized tests as a global norm is well-established, with their quality and efficacy widely acknowledged. However, insights from our survey of lecturers and students at technical universities in Odessa, Ukraine, and Ariel, Israel, align with findings by Tonya R. Moon, Catherine M. Brighton [7], Nathan R. Kuncel and Sarah A. Hezlett [3], and Rawlusyk P.E. [8]. These studies advocate for the integrated application of both standardized and non-standardized tests to achieve comprehensive knowledge assessment, capturing acquired competencies, skills, and the intellectual and creative capacities of students. Achieving a balanced application of these assessment methods is crucial for evaluating educational outcomes effectively.

Echoing Peg Tyre's [10] perspective, we concur that non-standardized tests devised by university lecturers within their respective disciplines should reflect the unique characteristics of the institution and the student's knowledge levels. This raises concerns about the quality of non-standardized tests, which are typically developed by educators without extensive expertise in test theory and methodology and lack the refinement process that pilot testing offers, which is essential for enhancing test quality based on established criteria.

Oleg Vynnychuk [11] highlights potential issues with the exclusive use of non-standardized tests for student knowledge assessment, including question relevance, subjective grading (leading to increased student stress and anxiety), unverified testing durations diminishing quality, absence of feedback, and heightened risks of plagiarism and cheating. Vynnychuk suggests that tests (both ST and NST) should not be the sole means of knowledge evaluation but should be complemented by creative and professionally oriented tasks, analytical writing, and other authentic assessment strategies.

Despite the growing implementation of computer-based testing technologies in the educational and scientific landscape [20] aimed at ensuring test regulation and objectivity, concerns about knowledge assessment quality persist. Often, the phrasing of questions may lead to intuitive guessing of answers, undermining assessment quality. The cumulative evidence from analyzed studies underscores the effectiveness of testing as a knowledge control method when tests are thoughtfully designed and applied alongside practical, skill-oriented tasks, adhering to rigorous test quality assessment criteria.

## 7. Conclusions

The literature review has delineated various criteria essential for ensuring the quality of testing, which may shift based on the unique objectives and conditions of each testing scenario. For this study, we identified and focused on the following test quality criteria: "efficacy", "usability", "reliability", "objectivity", "accuracy", and "relevance".

To explore the dynamics between the quality of knowledge assessment and the methods employed for such evaluations from the perspective of lecturers and students in technical fields, we conducted a survey. This survey involved rating both standardized tests (ST) and non-standardized tests (NST) on a 5-point scale across the identified quality criteria and providing rationales for their choices.

The survey results showed a clear preference for ST as the primary method for monitoring and assessing knowledge. Respondents valued ST for its ability to swiftly evaluate knowledge on specific topics, ensure equal testing conditions, and offer transparent assessment criteria. STs were deemed convenient, reliable, and objective, thanks to clear instructions, explicit assessment criteria, standardized procedures, and independence from the test compiler's expertise. Additionally, STs were noted for their accurate knowledge assessment capabilities and feedback potential. However, it was also mentioned that STs might only assess superficial knowledge and might not effectively evaluate practical skills, creative thinking, or educational competencies.

Despite receiving favorable evaluations across all criteria, respondents also highlighted the underrepresentation of STs in technical disciplines and noted the emotional challenges associated with test-taking, such as anxiety.

NSTs, while scoring slightly lower than STs, were still positively regarded by teachers and students. Their effectiveness was somewhat diminished due to challenges in presenting educational material in a structured and logical manner. NSTs were often criticized for being overloaded with answer options and lacking a regulated administration procedure, impacting their usability. Their reliability and objectivity were also questioned, particularly by students, who noted potential biases due to the non-standardized nature of test creation and administration. Accuracy issues were attributed to vague question formulations and concepts, potentially leading to assessment biases.

However, the relevance of NSTs was rated highly, attributed to the test compilers' (usually the course instructors) intimate knowledge of the curriculum, allowing for a nuanced inclusion of material based on its significance and complexity.

The feedback suggests a need to enhance the competency of NST compilers in test theory and methodology, advocating for the adoption of standardized methods in test construction, administration, and evaluation to address NST shortcomings.

Conclusively, the study reveals a marked preference for standardized testing methods among both lecturers and students for assessing knowledge, notwithstanding a marginal IQC difference between ST (0.72) and NST (0.64). The consensus on the efficacy of standardized assessment methods coexists with a recognized necessity for NSTs, which, when well-prepared and properly administered, can offer comprehensive knowledge assessments tailored to the unique requirements of each university discipline. Therefore, integrating both testing methodologies is recommended for optimal knowledge evaluation in higher education contexts.

**Author Contributions:** Conceptualization, N.D.; methodology, A.G. and O.K.; validation N.D., A.G. and O.K.; formal analysis, N.D.; investigation, N.D., A.G. and O.K.; resources, A.G. and O.K.; data curation, N.D.; writing—original draft preparation, A.G. and O.K.; writing—review and editing, N.D.; visualization, A.G. and O.K.; supervision, N.D.; project administration, N.D. All authors have read and agreed to the published version of the manuscript.

**Funding:** This research received no external funding.

**Institutional Review Board Statement:** Not relevant.

**Informed Consent Statement:** Not applicable.

**Data Availability Statement:** The original contributions presented in the study are included in the article.

**Conflicts of Interest:** The authors declare no conflict of interest.

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
