# Peer review of "Comparative Analysis of Knowledge Control and Evaluation Methods in Higher Education"

_education, doi:10.3390/educsci14050505_

Round 1

Reviewer 1 Report

Comments and Suggestions for Authors

The presented study deals with a relevant and exciting topic, is logically structured and presents the results comprehensibly. I also appreciate the international context of the study, which combines results from several countries. In this respect, the study is valuable and exciting.

Nevertheless, I would like to make a few critical comments:

1. Graphic design - please do not use the 3D effect for the graphs. Please work on the overall graphic design of the manuscript.

2. Citation standard is not followed, none of the studies have DOI.

3. It is impossible to base a study on working with "Avanesov" as the only relevant source. The text comes across (and I don't know who the authors are) as an attempt to increase the relevance or citation of a particular author from (mostly) mediocre sources.

4. The overall literature work does not meet the journal's standard. It is a small number of studies, dominated by authors (probably) from Russia or Ukraine, and lacks embedding in prestigious journals and their context. Textbooks or teaching texts are cited, which is unacceptable, at least not to this extent. If the text is to be accepted, the handling of literature needs to be fundamentally changed. I believe it is impossible to write a scholarly text with such an annotated apparatus, and it leads to the question of how relevant the topic is outside the countries analysed and what discourse on the topic prevails in the literature. It is not at all clear whether the results are in any way new or interesting.

5. Some references in the text do not have the year of publication.

6. I would not use "." or ":" for headings.

7. Please improve the description of what precisely the comparison of standardized and non-standardized tests looked like.

8. Why is a scale used: Yes - No - it is difficult to answer ? Is it valid? Its choice is not justified in the text.

9. Would it be possible to include the numbers of individual respondents from specific universities in the table? Are there any cross-cultural differences?

10. Although I am not a native speaker, I judge the language of the study to be "non-English", obviously combined (at least in some places) with machine translation in a way that is not entirely comprehensible.

Reviewer 2 Report

Comments and Suggestions for Authors

Well done!

Author Response

Please seethe attachment

Reviewer 3 Report

Comments and Suggestions for Authors

The literature review leaves out important work in this area.

Perhaps the description of standardised and non-standardised tests in the literature review should be in the research methods. It is also important that the authors justify the use of these tests.

The application of surveys and interviews to students and teachers should also be duly explained in the research methods. The results state the number of students and teachers surveyed, but nothing is known about the criteria behind the number of students and teachers surveyed. Only that they are from 4 universities.

The discussion of the results should be improved. It is very reductive.

Overall, the study is good enough to be published and should be improved.
